# Studying regional low-carbon development: A case study of Sichuan Province in China

**Genjin Sun**[1]*, **Rui Gao**[1], **Ying Liu**[1], **Yanxiu Liu**[2], **Cuilan Li**[3]

**1** School of Business and Tourism, Sichuan Agricultural University, Chengdu, China, **2** Research Institute of Economics and Management, Southwest University of Finance and Economics, Chengdu, Sichuan, China, **3** School of Economics and Trade, Guangdong University of Foreign Studies, Guangzhou, China

* 41406@sicau.edu.cn

## Abstract

The unavoidable option for socially sustainable development is a low-carbon economy. One of the essential steps for China to attain high-quality development is reducing carbon emissions. It is necessary to realize low-carbon development in Sichuan, as it is not only an important economic zone but also an ecological protected area. The concurrent relationship among energy consumption, carbon emissions, and economic growth was examined in this study using the Tapio decoupling indicator, and the factors affecting energy consumption and carbon emissions in Sichuan were broken down using the logarithmic mean Divisia indicator (LMDI). The findings demonstrate a fundamental relative decoupling relationship between Sichuan's energy use and carbon emissions. Analysis of energy consumption and carbon emissions in Sichuan Province from 2005 to 2020 shows distinct patterns. From 2005 to 2012, in 2014, and from 2016 to 2020, the relationship between energy use and carbon emissions was relatively decoupled, with decoupling values ranging between 0 and 1. Absolute decoupling occurred in specific years: 2010, from 2013 to 2018, and in 2020. These periods are characterized by economic growth alongside reductions in carbon emissions. Factors affecting energy consumption and carbon emissions were consistently analyzed, showing similar impacts throughout the study periods. We find that population and economic growth are the main driving forces of these effects. The effects of energy intensity and industrial structure mainly play restraining roles, and the latter has a slightly weaker effect than the former.

**Data Availability Statement:** The data for this study is available at https://www.openicpsr.org/openicpsr/workspace?goToPath=/openicpsr/200121&goToLevel=project.

## 1. Introduction

The greenhouse effect, significantly driven by human activities, has emerged as the principal contributor to global warming over the past century, with carbon dioxide emissions at the forefront [1]. This relentless warming trend poses a formidable challenge to the sustainable development of societies worldwide [2, 3]. As main carriers of human activities, the cities are deviating from sustainable development goals under the effect of urban heat islands (UHIs) [4]. It makes sense to reduce temperatures, enhance biodiversity and sequester carbon by optimizing city planning and development strategies considering urban climate, especially

**Funding:** This work was supported by Humanities and Social Sciences Research Planning Program of China's Ministry of Education (No.23YJA790070), awarded to GS; Li Bing Research Center (No. LBYJ2021-006), awarded to GS; Research Center of Tuojiang River Basin High-quality Development (No. TJGZL2021-14), awarded to GS; Chengdu Research Center for Integration into Dual - Circulation Development Pattern and Sichuan Research Center for Integration into Dual - Circulation Development Pattern(No. SXHY2023008), awarded to GS. The funders had no role in study design, data collection and analysis, decision to publish, or preparation of the manuscript.

**Competing interests:** The authors have declared that no competing interests exist.

expanding urban green spaces [5, 6], but the fundamental measure is to reduce carbon emissions at the source. In response, nations globally have initiated national programs aimed at curbing carbon emissions. Amidst this backdrop, China, witnessing a surge in carbon emissions due to escalating energy consumption and an inefficient energy structure, has emerged as the world's leading carbon emitter, accounting for 30.9% of global emissions in 2021 [7, 8]. In a significant move, China announced its ambitions for achieving a "carbon peak" by 2030 and "carbon neutrality" by 2060, with a commitment to a gradual decline in carbon dioxide emissions post-2030. The strategic adjustment of its energy consumption structure and the reduction in energy consumption intensity underscore China's dedication to fostering a low-carbon economy.

Sichuan Province is not only an ecological protected area but also an important economic zone located in southwest China (97°21′E-108°12E′, 26°03′N -34°19 N), with a total area of 486,00 km$^2$ Sichuan Province is located on the Sichuan-Yunnan ecological divide between the Tibetan Plateau and the Lesser Plateau, with greatly different landscapes and vastly complex terrain. It serves as a major center for the conservation of biodiversity worldwide in addition to being a key water supply area for the upper reaches of the Yangtze River and a key recharge area for the upper reaches of the Yellow River. It is a crucial element of the nation's plan for environmental safety. Meanwhile, the economy of Sichuan Province has grown rapidly in recent decades, according to the National Bureau of Statistics. Sichuan's gross domestic product (GDP) in 2021 was 5.39 trillion yuan, ranking sixth in China. However, energy consumption has been growing rapidly during the economic development process of Sichuan Province. Its average growth rate over the last five years has been approximately 2.96%. The total energy consumption in Sichuan Province will reach 230 million tonnes of standard coal in 2021. Of the total energy consumption, 25.9% will be coal combustion consumption, 17% will be oil fuel consumption, 16.7% will be natural gas consumption, and 40.4% will be primary electricity and other energy consumption (as shown in Fig 1). The province's non-fossil energy consumption accounts for 39.5% of total energy consumption, and thus it more than 20 percentage points higher than the country as a whole, ranking first in China. These data show that Sichuan Province has been experiencing a high level of energy consumption and an irrational energy mix during the economic growth period. In terms of carbon emissions, Sichuan Province showed a fluctuating upward trend from 2005 to 2012, followed by a slowly declining period. However, it was still 78.9086 million tonnes by 2020 (as shown in Fig 2). Therefore, to support the growth of a low-carbon economy in Sichuan Province, it is of utmost importance to research the interactions between the use of energy, the release of carbon dioxide, and economic development. The findings of our study will provide useful inspiration for the development of low-carbon economies in ecologically protected areas around the world.

Decoupling analysis is frequently used to examine the relationship between carbon emissions and economic growth and decompose the factors affecting carbon emissions [9], even though it was initially used to describe the relationship between energy consumption and economic growth [10–12]. In terms of the subject, some scholars have highlighted this topic with a certain country [13–16], or a certain region [17–19]. From the perspective of methodology, Tapio decoupled elasticity coefficient theory quantitatively analyses the situation where the pollution emissions growth rate changes as the economic growth rate changes, utilizing the decoupling factors proposed by the Organization for Economic Co-operation and Development (OECD) [20, 21]. Meanwhile, index decomposition analysis (IDA), structural decomposition analysis (SDA), and production-theory decomposition analysis (PDA) are the main decomposition techniques for carbon emission components. LMDI is widely used in factor analysis and is regarded as a preferred IDA decomposition method [22]. The existing studies show that although there are many academic studies on the relationship between energy

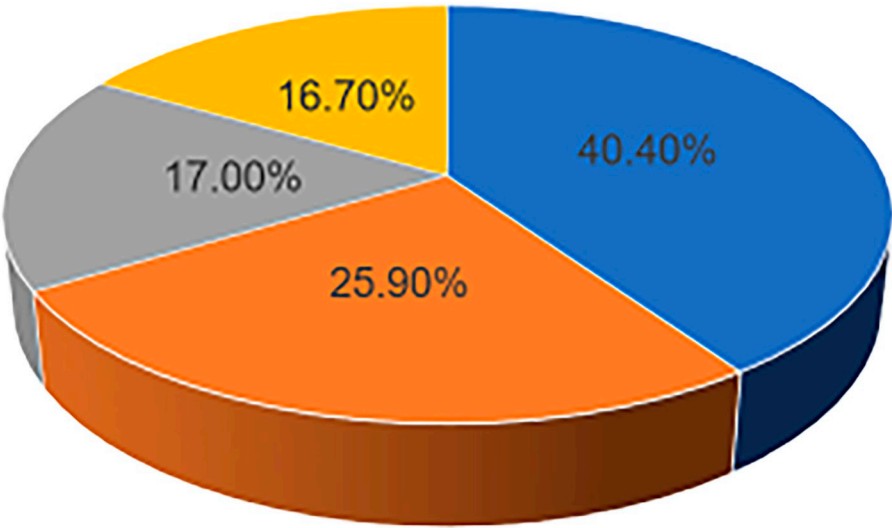

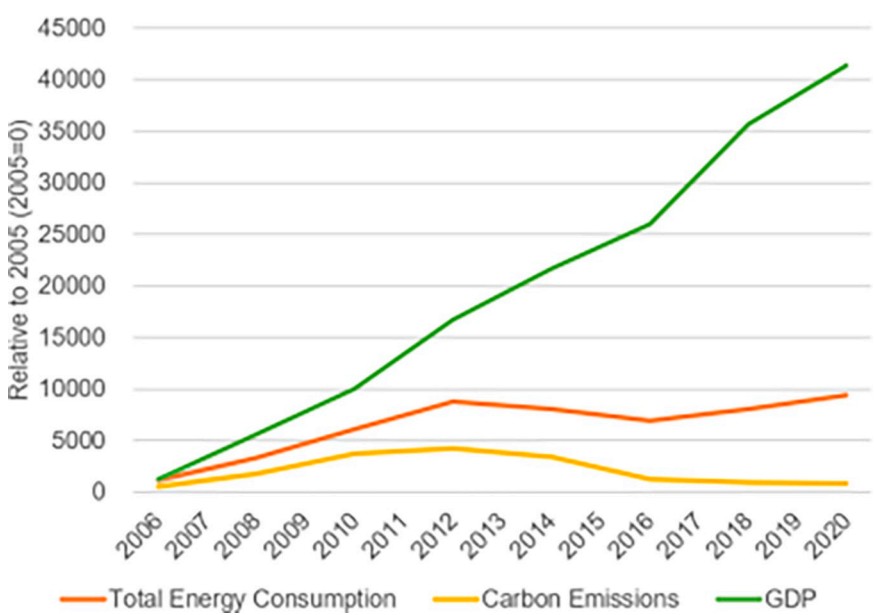

**Fig 1. The energy consumption structure of Sichuan Province (2021).**

**Fig 2. Total energy consumption, carbon emissions, and GDP in Sichuan Province (2006–2020).**

consumption and economic growth and on the distribution of factors affecting $CO_2$ emissions, the conclusions are not consistent due to different research topics or different research methods.

Research directions within the realm of environmental economics and sustainable development are diverse, unveiling the intricate interplay between economic activities and environmental outcomes. A pivotal focus among current research trajectories is the investigation of the relationship between economic growth and environmental degradation, often contextualized within the framework of the Environmental Kuznets Curve (EKC) hypothesis [23–25]. This hypothesis posits an inverted U-shaped relationship between environmental pollutants and per capita income, suggesting that pollution levels increase with economic growth up to a certain threshold, beyond which they begin to decline. Through the application of econometric analyses, incorporating time-series, cross-sectional, and panel data, scholars assess the dynamics between economic development and various forms of environmental impacts. Sustainable energy transition emerges as another central theme, emphasizing the shift from fossil fuel-based energy systems to those relying on renewable sources and exhibiting lower carbon intensity. Research in this domain typically leverages scenario analysis and modelling techniques, such as the Long-range Energy Alternatives Planning (LEAP) system [26, 27] or Integrated Assessment Models (IAMs) [28–30], to explore future energy pathways and their implications for climate change mitigation and energy security. Concurrently, resource efficiency and the circular economy are increasingly garnering attention. Studies in this area apply methodologies like Life Cycle Assessment (LCA) [31, 32], Material Flow Analysis (MFA) [33], and Input-Output Analysis (IOA) [34, 35] to comprehend the environmental impacts of products, services, and economic sectors throughout their entire life cycles. These methods aid in identifying opportunities for enhancing efficiency and implementing circular economy principles. Another significant research direction involves exploring the concept of decoupling, which refers to the capacity for an economy to grow without a corresponding increase in environmental pressure. Studies on decoupling utilize indicators such as the Tapio decoupling indicator to analyze the extent of economic growth relative to environmental degradation. This methodology is instrumental in evaluating the effectiveness of sustainable development policies.

However, these research endeavors are not without limitations. Firstly, many studies predominantly focus on the macro-level, with insufficient analysis at regional or provincial scales, overlooking inter-regional disparities. Existing research often emphasizes a single indicator or methodology, lacking attempts to comprehensively assess and analyze using a variety of tools and methods. Regarding strategies and measures proposed for effectively achieving a balanced and sustainable development between energy consumption, carbon emissions, and economic growth, further in-depth exploration and validation are necessary. Moreover, there is little research on the decoupling of energy use and carbon emissions from economic growth or, more specifically, on the decoupling of energy use and carbon emissions in Sichuan Province. Thus, our goal is to close this gap in the literature. The study contributes to the current work on the development of the regional low-carbon economy in three major aspects. To the best of our knowledge, this study sets precedence by analyzing the factors affecting energy consumption and carbon emissions and decoupling the connection between energy consumption, carbon emissions, and economic growth. The research field of regional low-carbon economic development will be expanded in this study. Second, this paper takes Sichuan Province as the research object. The special feature of Sichuan Province is that it is not only an important economic zone but also an ecological protected area. Many things are different from other regions in terms of balancing economic development and environmental protection. Therefore, our study enriches the case studies of regional sustainable development. Finally, as far as we know,

this is the first study to find that the population and economic growth are the main driving forces for low-carbon economic development in Sichuan Province, and the effects of energy intensity and industrial structure mainly play restraining roles. The findings will not only offer a direction for Sichuan Province in terms of its pursuit of a low-carbon economy, but they will also provide a piece of advice for other environmentally sensitive emerging nations and regions. Besides, this study aims to delve into the concurrent relationships between energy consumption, carbon emissions, and economic growth, particularly focusing on the specific case of Sichuan Province in China. Based on the aforementioned analysis, targeted policy recommendations are proposed to promote harmonious development between the economy and the environment in Sichuan Province and beyond while simultaneously reducing energy consumption and carbon emissions. It is hoped that this research will provide valuable insights and strategies for achieving sustainable development at both regional and global levels.

The remaining sections of this study are arranged as follows. Section 2 provides the methodology. The results of the decoupling analysis on the consumption of energy, emissions of carbon dioxide, economic growth, and the decomposition of factors are described in Section 3. We critically discuss the results and put forward policy recommendations and scope for future research in Section 4. Section 5 concluded our study.

## 2. Methods

In order to analyze the development of the regional low-carbon economy in Sichuan Province, the Tapio decoupling index is adapted to test the decoupling relationship between energy use, carbon emissions, and economic growth, and the LMDI is used to decompose the variables affecting energy consumption and carbon emissions. The former can visualize the degree of decoupling between energy use, carbon emissions, and economic growth, and the latter can analyze the main factors affecting energy consumption and carbon emissions in Sichuan Province, as well as the degree of influence of each factor. The methods are introduced as follows.

### 2.1 Tapio decoupling index

The Tapio elasticity coefficient method was first created as a decoupling indicator using the concept of decoupling elasticity in the study of the decoupling of carbon dioxide emissions from the transportation sector and economic growth in the European Union and Finland [20]. We use the decoupling model to analyze the decoupling relationship between energy consumption, carbon emissions, and economic growth. The calculation formula of the decoupling index is as follows:

$$D_{E, Y} = \frac{\Delta E'}{\Delta Y'} = \frac{\Delta E / E}{\Delta Y / Y} \tag{1}$$

$$D_{C, Y} = \frac{\Delta C'}{\Delta Y'} = \frac{\Delta C / C}{\Delta Y / Y} \tag{2}$$

$D_{E,Y}$ and $D_{C,Y}$ are the elasticities of change in energy consumption and carbon emissions concerning GDP growth, that is, the change in energy consumption and carbon emissions per percentage point change in GDP. $\Delta E'$, $\Delta E$, and E are the rate of change, the amount of change and the initial value of energy consumption, respectively. $\Delta C'$, $\Delta C$, and C are the rate of change, the amount of change and the initial value of carbon emissions, respectively. $\Delta Y'$, $\Delta Y$, and Y are the rate of change, the amount of growth and the initial value of GDP, respectively.

The decoupling states of Tapio's decoupling model can be categorized into eight categories: strong decoupling, weak decoupling, recessive decoupling, strong negative decoupling,

**Table 1. Criteria for judging the degree of decoupling of economic output from carbon emissions.**

| Trends | $\Delta C'$ | $\Delta Y'$ | $D_{C,Y}$ | Type |
|---|---|---|---|---|
| Decoupling | $(-\infty, 0)$ | $(0, +\infty)$ | $(-\infty, 0)$ | Strong Decoupling |
| | $(0, +\infty)$ | $(0, +\infty)$ | $(0, 0.8)$ | Weak Decoupling |
| | $(-\infty, 0)$ | $(-\infty, 0)$ | $(1.2, +\infty)$ | Recessive Decoupling |
| Negative Decoupling | $(0, +\infty)$ | $(-\infty, 0)$ | $(-\infty, 0)$ | Strong Negative Decoupling |
| | $(0, +\infty)$ | $(0, +\infty)$ | $(1.2, +\infty)$ | Expansive Negative Decoupling |
| | $(-\infty, 0)$ | $(-\infty, 0)$ | $(0, 0.8)$ | Weak Negative Decoupling |
| Coupling | $(-\infty, 0)$ | $(-\infty, 0)$ | $(0.8, 1.2)$ | Recessive Coupling |
| | $(0, +\infty)$ | $(0, +\infty)$ | $(0.8, 1.2)$ | Expansive Coupling |

expansive negative decoupling, weak negative decoupling, recessive Coupling, and expansive Coupling, as shown in Table 1. Among them, strong decoupling is the most desirable state, and strong negative decoupling is the least desirable state.

## 2.2 LMDI model construction

Decomposition analysis is an excellent way to visualize economic growth, and the LMDI is highly adaptable compared to other methods [36]. LMDI allows us to precisely quantify the role of the level of technology, industrial structure, inflation, and population growth in terms of economic growth over several years. The results can be used to summarize or confirm economic development patterns and provide guidance for future investment or reform policies. There are two main approaches to decomposition models: IDA (INDEX decomposition analysis) and SDA (structural decomposition analysis). The former is a progressive and improved version of the latter. The latter can be divided into two methods: the Laspeyres index decomposition method and the Divisia index decomposition method [37]. As one of the most frequently used carbon emission factor decomposition techniques, LMDI can produce a fair factor decomposition, and the findings do not include inexplicable residual terms. The number of decomposition factors mainly includes four-factor and five-factor decomposition. For example, Wang et al. (2005) used LMDI to decompose China's carbon emissions into four factors: population, GDP per capita, energy intensity, and energy consumption structure [38]. Ma et al. (2008) decomposed the change in China's carbon dioxide emissions into the effects of population, GDP per capita, carbon-free energy incorporation, biomass substitution, and fossil fuels [39]. The calculation formula for LMDI is given below.

$$C = \sum_{j=1}^{n} C_i = \sum_{j=1}^{n} \frac{E_i}{GDP_i} \times \frac{GDP_i}{GDP} \times \frac{GDP}{POP} \times GDP = \sum_{j=1}^{n} I_i \times S_i \times P \times GDP \qquad (3)$$

In the above equation, C denotes regional carbon emissions, n denotes the number of industrial sectors, $C_i$ denotes carbon emissions from sector i or industry, $E_i$ denotes energy use in sector i, $GDP_i$ denotes gross domestic product in sector i, and POP denotes population. $I_i = \frac{E_i}{GDP_i}$ denotes energy use efficiency, that is, energy consumed per unit of GDP. $S_i = \frac{GDP_i}{GDP}$ denotes the industrial structure, that is, the proportion of output in sector i to total output. $P = \frac{GDP}{POP}$ denotes GDP per capita.

According to the full decomposition model, the change in carbon emissions, ($\triangle$C), can be differently decomposed into energy efficiency (intensity) impact ($\triangle$I), industrial structure impact ($\triangle$S), population size impact ($\triangle$P) and economic growth impact ($\Delta$GDP) from the

base period "0" to the period "t". Its calculation formula is as follows.

$$\Delta C = C^t - C^0 = \sum_{i=1}^{n} I_i^t \times S_i^t \times P^t \times GDP^t - \sum_{i=1}^{n} I_i^0 \times S_i^0 \times P^0 \times GDP^0$$

$$= \Delta I + \Delta S + \Delta P + \Delta GDP \quad (4)$$

The decomposition of the influence factor on the right-hand side of the above equation can be expressed as follows.

$$\Delta I = \sum_i \frac{C_i^t - C_i^o}{\ln C_i^t - \ln C_i^0} \times \ln(\frac{I_i^t}{I_i^0}) \quad (5)$$

$$\Delta S = \sum_i \frac{C_i^t - C_i^o}{\ln C_i^t - \ln C_i^0} \times \ln(\frac{S_i^t}{S_i^0}) \quad (6)$$

$$\Delta P = \sum_i \frac{C_i^t - C_i^o}{\ln C_i^t - \ln C_i^0} \times \ln(\frac{P^t}{P^0}) \quad (7)$$

$$\Delta GDP = \sum_i \frac{C_i^t - C_i^o}{\ln C_i^t - \ln C_i^0} \times \ln(\frac{GDP_i^t}{GDP_i^0}) \quad (8)$$

## 2.3 Data sources

The information on population, GDP, total primary energy output, and total energy consumption used in this study is obtained from the Sichuan Statistical Yearbook. The primary industries are those related to farming, forestry, animal husbandry, and fishing; the secondary industries are those related to industry and construction; and the tertiary industries are those related to transportation, storage, postal services, wholesale, retail, lodging, and catering. The material balance technique is used in the computation of carbon dioxide emissions, and the data include information on the consumption of coal, oil, and natural gas. Because the energy consumption obtained in the statistical yearbook has been converted into standard coal data, it needs to be converted back into physical quantity. We use the conversion coefficient of standard coal published by the National Development and Reform Commission to calculate the physical quantity of various materials used in primary energy consumption. The carbon emission factors are calculated using information from the US Department of Energy (DOE), the statistical agency Energy Information Administration (EIA), the Japan Energy Research Institute (JERI), and the National Science Council Climate Project. The average values from these well-known institutions listed above are shown in Table 2. This study did not use any kind of human participants or human data, which requires any kind of approval.

## 3. Results

### 3.1 Decoupling analysis

Fig 3 illustrates the decoupling of energy consumption, carbon footprint, and economic growth in Sichuan Province. Despite a weak decoupling, Sichuan Province's energy consumption and economic growth exhibit overall strength. The decoupling value of energy consumption typically ranges between 0 and 0.8, signifying a weak decoupling state. However, in 2013 and 2015, it decreased to -0.6104 and -1.5752, respectively, indicating a strong decoupling

**Table 2. Carbon emission factors for energy consumption.**

| Research Institutions | Coal | Oil | Natural gas |
|---|---|---|---|
| DOE/EIA | 0.7020 | 0.4780 | 0.3890 |
| Japan Energy Economics Institute | 0.7560 | 0.5860 | 0.4490 |
| Energy Research Institute of the National Development and Reform Commission | 0.7476 | 0.5825 | 0.4435 |
| Chinese Academy of Engineering | 0.6800 | 0.5400 | 0.4100 |
| NEPA Greenhouse Gas Control Project | 0.7480 | 0.5830 | 0.4440 |
| National Science Council Climate Change Project | 0.7260 | 0.5830 | 0.4090 |
| Average | 0.7266 | 0.5588 | 0.4241 |

state. This suggests that while energy consumption in Sichuan Province increased alongside GDP during this period, it did so at a slower rate than GDP growth, aligning the expansion of GDP with the decrease in energy consumption.

The decoupling effect value of Sichuan Province's carbon emissions fluctuates significantly, ranging from 1.2 to -4, with notable fluctuations. Specifically, carbon emissions in 2006–2008, 2010, 2012, and 2019 exhibited weak decoupling states, where carbon emissions increased alongside GDP growth but at a slower rate. In contrast, carbon emissions in 2011, 2013–2018, and 2020 demonstrated strong decoupling states, indicating a decrease in carbon emissions alongside GDP growth. The decoupling value in 2009 was 1.1104, reflecting expansive decoupling. It is important to highlight that the decoupling index for carbon emissions decreased to -3.559 in 2015.

## 3.2 Decomposition of factors affecting changes in energy consumption

According to the LMDI factor decomposition of energy consumption (Fig 4), the average contributions of population size and economic growth are positive, indicating that population and economic growth are driving factors for the sustained growth of energy consumption demand. The effect of energy intensity is negative, suggesting that the decline in energy consumption intensity has slowed down with the growth of total energy consumption. The impact of industrial structure has been negative since 2011. From 2005 to 2010, the industrial structure had a

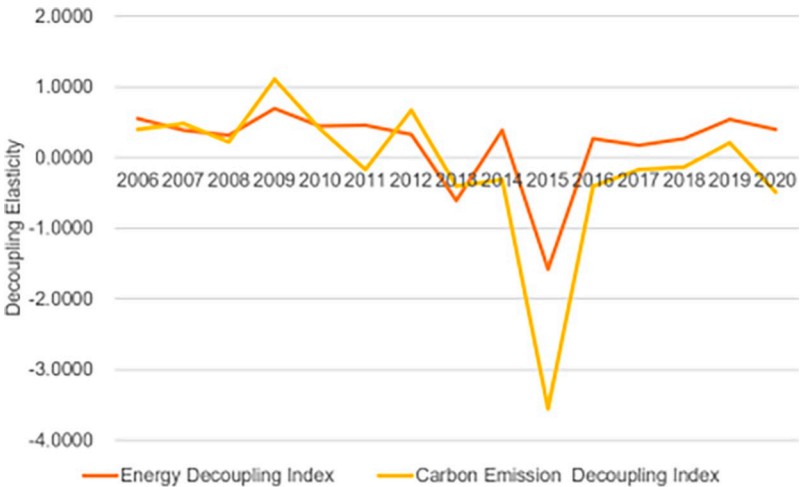

**Fig 3. Decoupling of energy consumption and carbon emission changes from economic growth in Sichuan Province (2005–2020).**

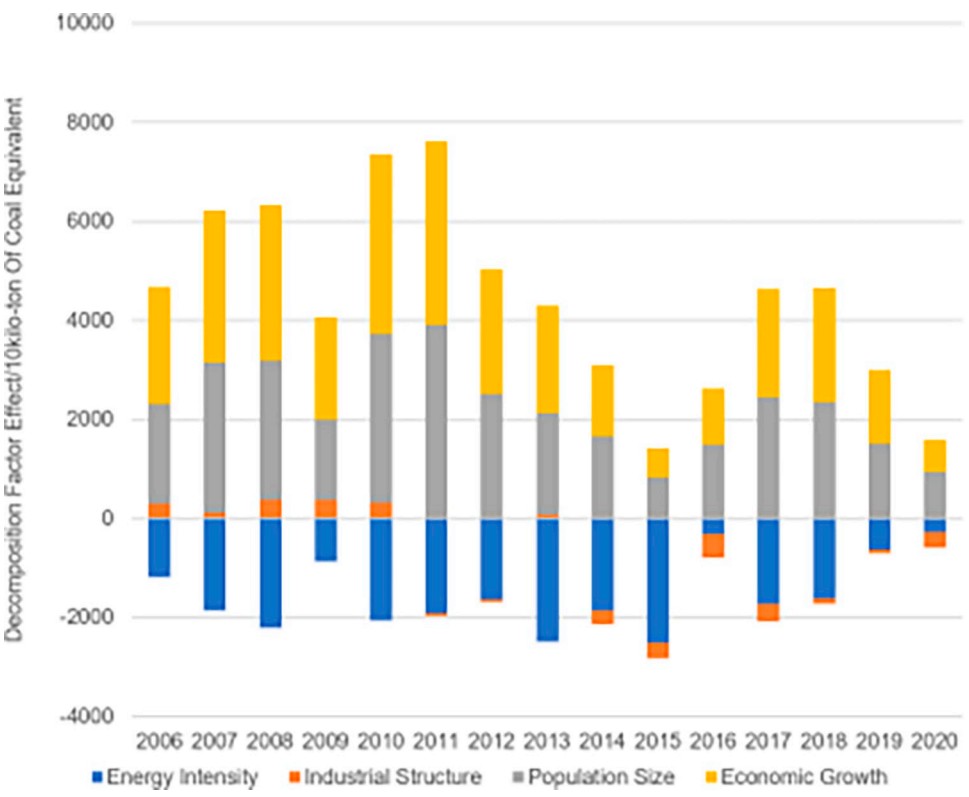

**Fig 4. Breakdown of factors influencing interannual changes in energy consumption in Sichuan Province (2005–2020).**

positive driving effect on energy consumption, it shifted to a negative inhibitory effect after 2011.

### 3.3 Decomposition of factors affecting changes in carbon emissions

Fig 5 presents the effect values of energy intensity, industrial structure, population size, and economic growth on carbon emission changes in Sichuan Province. Overall, economic growth and population size in Sichuan Province from 2005 to 2020 have shown positive impacts, indicating their significant roles in the development of carbon emissions. Energy intensity consistently exhibits a negative effect, acting as a suppressor in carbon emission growth. The influence of industrial structure has been relatively minimal, playing a weak role in carbon emission changes.

## 4. Discussions

To gain more experience in promoting low-carbon economic development in the ecologically protected areas of developing countries, we will review and discuss the empirical results in this section.

### 4.1 Decoupling of energy consumption and influencing factors of energy consumption change

Overall, energy consumption and carbon emissions in Sichuan Province exhibit a mix of weak and strong decoupling from economic growth, showcasing a favorable trend. The study based

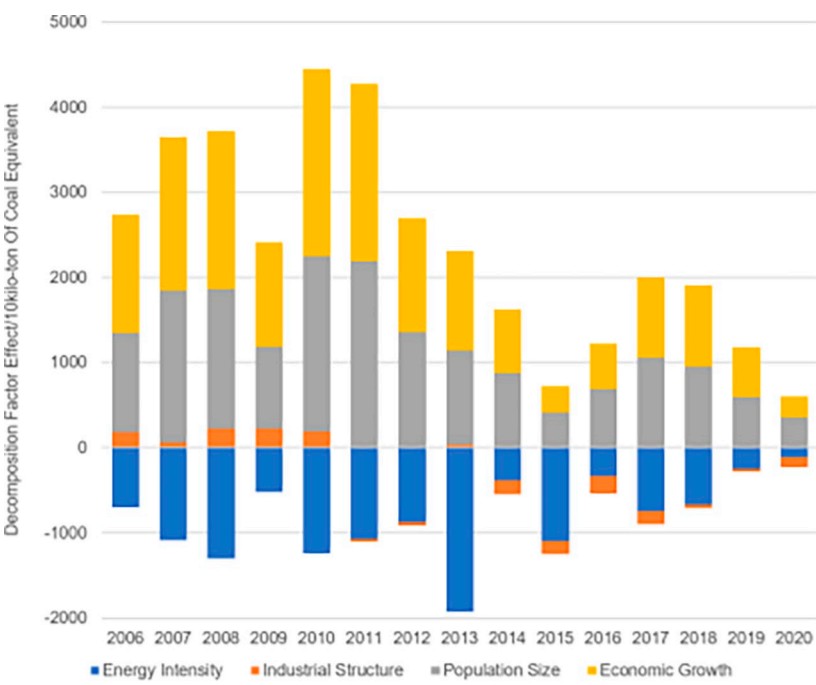

**Fig 5. Results of decomposition of factors influencing interannual changes in carbon emissions in Sichuan Province (2005–2020).**

on LMDI results reveals a consistent increase in energy demand driven by population and economic growth in Sichuan Province from 2005 to 2020. The effects of energy intensity and industrial structure have predominantly been negative until 2011, impeding the overall growth of energy consumption. However, post-2011, the industrial structure has played a significant role in boosting energy demand. These findings align with earlier research [40], indicating that the energy landscape in Sichuan Province is evolving under governmental and other influences.

These empirical findings are closely tied to Sichuan's enhanced policy framework for energy consumption and the stringent implementation of energy conservation plans during the Eleventh and Twelfth Five-Year Plans. In terms of primary energy production, the share of coal is decreasing while direct electricity and natural gas percentages in clean energy are rising. Crude oil shares remain stable at 0.1% to 0.2%. This shift is attributed to the aggressive development of Sichuan's hydropower sector during the Twelfth Five-Year Plan, culminating in peak installed capacity exceeding 12 million kilowatts in 2013. Sichuan Province leads in China with an installed hydropower capacity of 89.04 million KW by the end of 2021. The growth rates of natural gas consumption, primary electricity, and other clean energy sources surpass those of coal, reflecting a growing market for renewable energy and changing consumer preferences toward clean and low-carbon energy sources. Despite advancements in energy consumption and production, the challenge of rising energy demand persists.

In Sichuan Province, population dynamics primarily exert a positive driving force on carbon emissions, albeit at a reduced level in recent years. Demographic factors interact with economic, resource, and environmental factors, with urban expansion and infrastructure development contributing to increased energy consumption [41]. While the implementation of new fertility policies has commenced, strengthening the positive impact of population size on energy consumption remains a challenge. The slowdown in population growth post-

implementation of family planning policies has effectively managed the population's impact on energy consumption.

Fig 2 illustrates significant GDP growth in Sichuan Province, accompanied by a general upward trend in total energy consumption. Despite a decline in total energy consumption during 2012–2016, energy consumption notably increased from 2005 to 2012. While energy consumption fluctuated in subsequent years, with decreases in 2013 and 2015, the pattern aligns with shifts in non-clean energy consumption and efforts towards energy conservation, emission reduction, and a carbon-neutral economy. This underscores the correlation between economic expansion and energy utilization.

The average effect of industrial structure on carbon emissions resulting from energy consumption in Sichuan Province is -13.25%. Initially a positive driver of energy consumption from 2005 to 2010, the impact of industrial structures transitioned to an inhibitory effect after 2011. This trend mirrors Sichuan's industrial evolution, where the proportion of secondary industry initially rises before declining, leading to increased energy consumption per unit of GDP [42]. Similar conclusions have been drawn in previous studies [43], highlighting the evolving dynamics of industrial structure and its impact on energy consumption in the region.

Combining energy flows by sector and type for four time spans: 2005, 2010, 2015, and 2020, as shown in Figs 6–9. Based on these four graphs, it can be found that for the secondary sector in Sichuan Province, coal and electricity account for the largest share of energy consumption. The tertiary sector, including transport, storage, postal, wholesale, retail, accommodation, and tertiary industries, has a relatively fragmented primary energy consumption. The transport, storage, and postal sectors consume more oil due to the long distances involved in transport. In contrast, the household sector consumes mainly electricity. Among the three main sectors, the primary sector, mainly agriculture, forestry, and fishery, consumes less energy. The secondary sector, especially manufacturing and construction, generally suffers from overcapacity and very aggregated energy consumption, with significantly higher energy consumption than

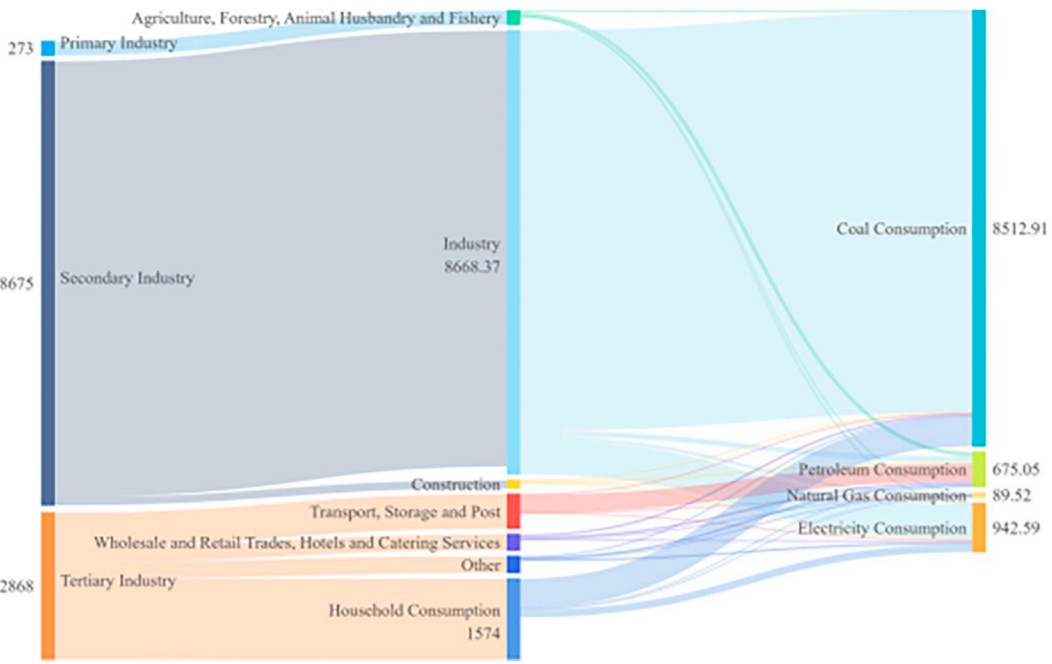

**Fig 6. Diagram of energy flows by industry and species in Sichuan Province (2005).**

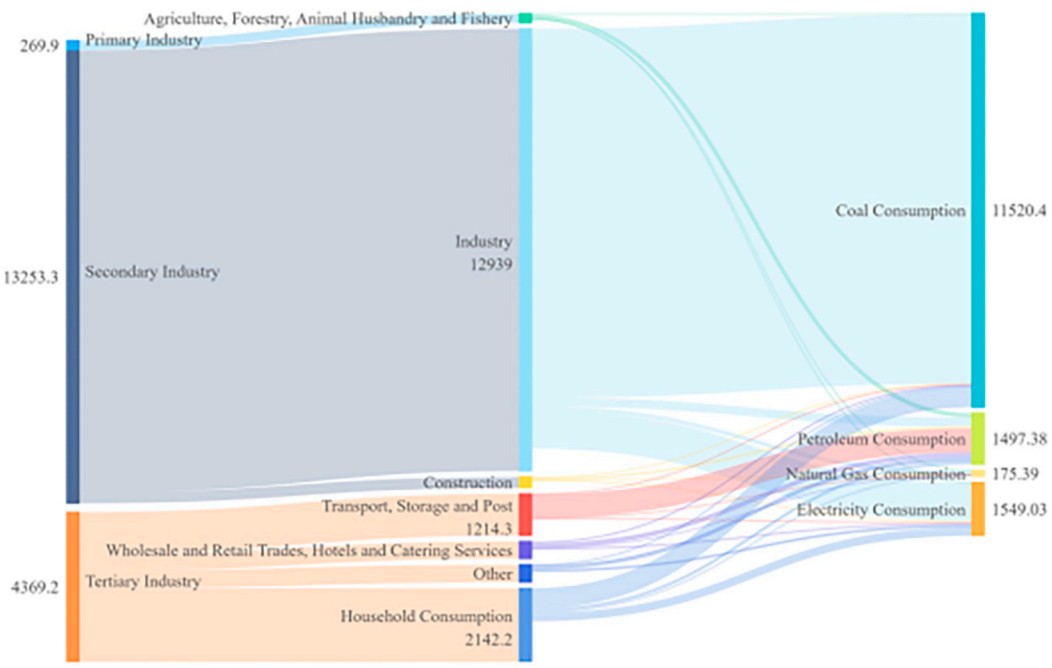

**Fig 7. Diagram of energy flows by industry and species in Sichuan Province (2010).**

the other sectors. During the 13th Five-Year Plan period, supply-side structural reform strongly contributed to the optimization and modernization of the industrial structure, increasing the role of industrial restructuring in curbing the growth of excessive energy consumption. Due to the dependence of the secondary sector on energy consumption, reducing

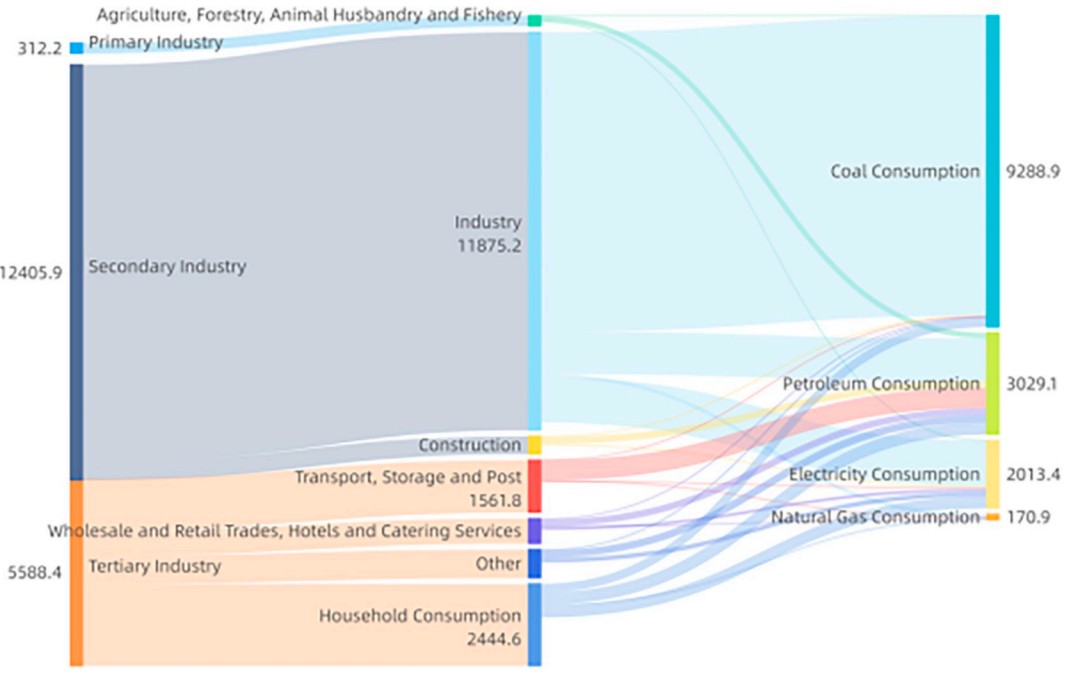

**Fig 8. Diagram of energy flows by sector and species in Sichuan Province (2015).**

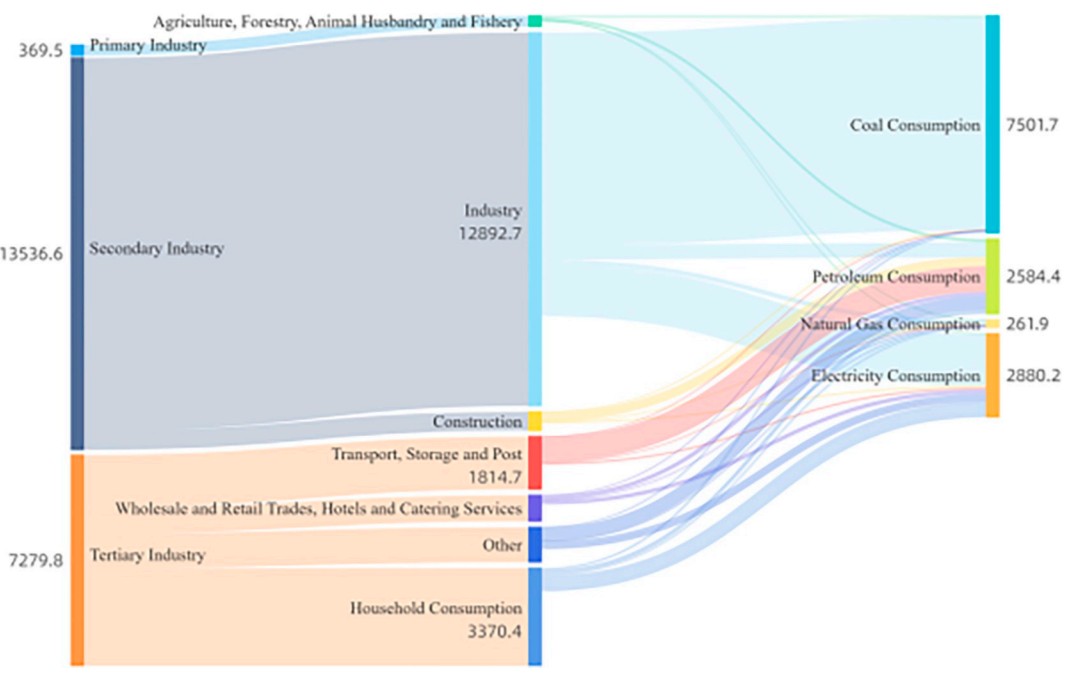

**Fig 9. Schematic diagram of energy flows by sector and species in Sichuan Province (2020).**

overall energy consumption will require further restructuring of the secondary sector and a significant reduction in energy demand from traditionally energy-intensive industries. Overall, there were no significant changes in energy flows across sectors.

As an indicator of energy consumption efficiency, energy consumption intensity can reflect the technical level to a certain extent [44]. The energy consumption intensity of Sichuan Province has maintained a downward trend from 2005–2020, from 1.6421 to 0.4359, and this trend is closely related to the consistent adherence to the promotion and application of low-carbon technologies [45]. Since 2020, energy consumption has increased in addition to the conventional statistics of coal, electricity, oil, and natural gas. Other types of energy are used for industrial and residential consumption in Sichuan Province, so the energy structure and variety have been optimized. In recent years, Sichuan Province has introduced several policies, such as the "Work Plan for Controlling Greenhouse Gas Emissions in Sichuan Province" and the "Action Plan for Energy Saving and Low Carbon Development in Sichuan Province (2014–2015)", to encourage more research and development on low carbon technologies. Meanwhile, particular actions on energy saving and emission reduction technologies are vigorously carried out to promote green development.

To establish a "National Clean Energy Demonstration Province," Sichuan Province should propose more reasonable planning for the energy sector to improve the efficiency of conventional energy consumption effectively. Some new types of energy technology should be accelerated, and a large amount of clean energy has been put into enterprise production and residents' lives [46].

## 4.2 Decoupling of carbon emissions and influencing factors of carbon emissions

Based on carbon emissions and economic growth estimates, the decoupling effect varies widely between 1.2 and 4 in Sichuan. An examination of the decoupling of carbon emissions and

economic growth in Sichuan Province and the factors influencing it shows a decoupling value of 1.1104 in 2009, indicating a strong correlation with economic growth. The growth rate of $CO_2$ emissions was higher than that of GDP, mainly due to the high coal consumption during these years. The higher the coal consumption is, the higher the energy consumption. Inefficient coal burning led to high emissions of carbon dioxide, soot, and dust, resulting in an increase of 10.99 million tonnes of carbon dioxide emissions in Sichuan.

Sichuan Province has responded positively to the "double decarbonization" objective in terms of policy during recent years [47]. They optimally tried their best to vigorously promote energy savings, reduce energy consumption, and increase carbon sinks. Meanwhile, low-carbon development is promoted through industrial systems, production methods, lifestyles, and consumption patterns.

As seen from the carbon emissions of the three fossil energy sources in Sichuan Province (Fig 10), raw coal carbon emissions dominate the total fossil energy emissions, and controlling coal consumption has always been the key to curbing the intensity of carbon emissions [48]. However, carbon emissions from crude oil and natural gas consumption in Sichuan Province are gradually increasing. Specifically, carbon emissions from raw coal, crude oil, and natural gas fluctuated from 2005 to 2010, with total carbon emissions rising and carbon intensity exceeding 0.5 tons per 10000 yuan. An analysis of the reasons behind this reveals that the period 2006–2010 was the period of the 11th Five-Year Plan, a critical period for building a moderately prosperous society, during which Sichuan Province focused on economic development and vigorously implemented the strategy of strengthening the province through industry, resulting in carbon emissions from significant fossil energy sources not being reasonably controlled. During 2012–2020, carbon emissions from raw coal gradually decreased from 70,042,787 tonnes in 2012 to 42,768,403 tonnes in 2020 due to the 12th and 13th Five-Year Plans. Sichuan Province has continued to promote energy conservation and emission reduction and actively built a regional carbon emission reduction mechanism, thus leading to a

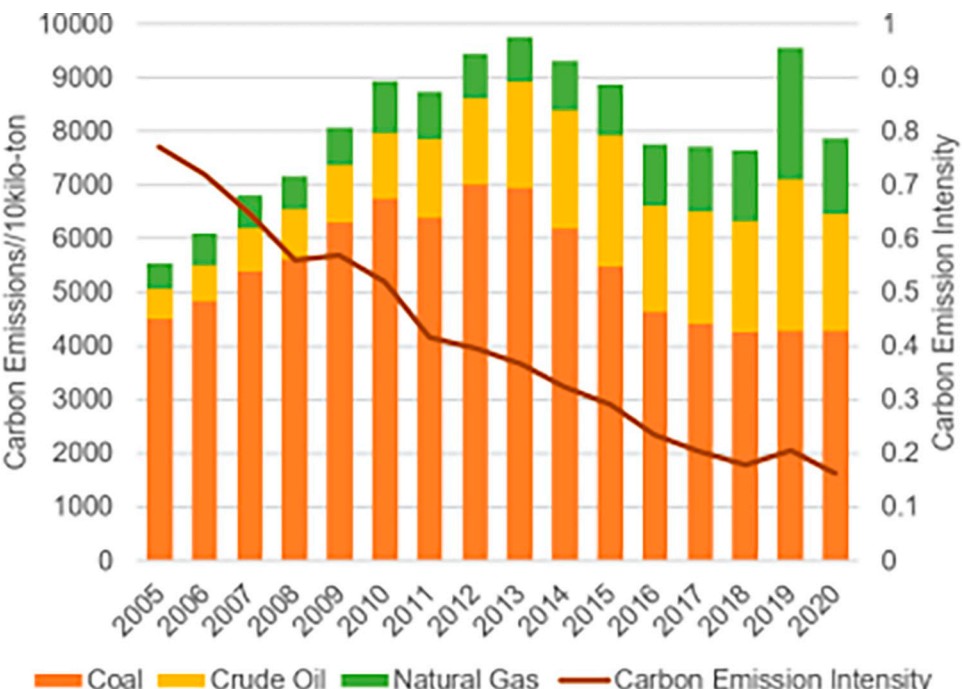

**Fig 10. Carbon emissions and carbon intensity of the three fossil energy sources in Sichuan Province (2005–2020).**

planned regulation of coal consumption and a slow decline in carbon emissions from raw coal. At the same time, the carbon emission intensity has dropped to below 0.5 tons per 10,000 Yuan and has shown a continuous downward trend. However, given the acceleration of economic development, Sichuan Province's energy consumption demand will not drop significantly shortly, and the energy conservation and emission reduction tests are still severe. From the perspective of the decomposition factors, it is evident that the driving effects of economic growth and population have significantly weakened after 2011, although economic growth and population have a positive impact on the development of carbon emissions. During the process, the effect of economic growth and population increased from 2083.7438 and 2193.5464 in 2011 to 254.2551 and 352.0754 in 2020, respectively. This is a positive outcome of Sichuan Province's response to national efforts to control greenhouse gas emissions and promote a low-carbon consumption pattern, production, and lifestyle. Additionally, with the further implementation of the "Chengdu-Chongqing twin-city economic circle" strategy, Sichuan Province will gradually transform and upgrade its economic development, adhere to the low-carbon development path, and continuously improve the quality of its economic development [49]. It is noteworthy that the economic growth and population size effect values increased again in 2017–2018, most likely due to the rapid economic growth rate and the lack of reasonable limits on carbon emissions in economic development. In terms of energy efficiency, the effect of energy efficiency remains negative throughout the statistical years. The energy efficiency effect had a minimum value of -1927.6002 in 2013, indicating that energy efficiency had the most inhibiting impact on carbon emission growth in 2013. Subsequently, the energy efficiency effect increased between 2014 and 2020, resulting in a decrease in the inhibiting effect of energy efficiency on carbon emissions growth, further highlighting the existing issues in recent years regarding energy use in Sichuan Province and the necessity to enhance the conversion and efficiency of energy use in Sichuan [50]. With a few exceptions, the industrial structure impact has a negative value from 2005 to 2020, largely preventing Sichuan Province's carbon emissions from increasing. The industrial structure effect particularly exhibited positive values in the early years of 2005–2010 and 2013, indicating that the industrial structure played a role in promoting the rise of greenhouse gases during this period. The secondary sector accounted for the majority of the province's total energy consumption during this period, while the primary and tertiary sectors accounted for a smaller percentage. This aligns with the findings of the national sample [51]. The industrial structure of Sichuan Province has a weak inhibitory influence on the rise of carbon emissions, as evidenced by the negative effect of industrial structure in the remaining years, with effect values ranging between -213 and -31.

## 4.3 Policy implications

In summary, in the context of Sichuan Province, the empirical findings reveal a complex relationship between energy use, carbon dioxide emissions, and economic growth. The findings underscore the effectiveness of "low carbon" strategies implemented since 2011 and the potential of industrial structure adjustments to limit the growth of carbon dioxide emissions. Theoretically, this study enriches the research content of regional economics by illustrating the intricate and dynamic relationship between economic growth, energy consumption, and carbon emissions through a detailed case study of Sichuan Province. In addition, this study contributes to an in-depth understanding of the mechanisms behind energy efficiency and industrial structure optimization, providing valuable insights for formulating sustainable development policies. More specifically. The following four recommendations are made for the future development of Sichuan Province, considering the "peak carbon" and "carbon neutral" targets and the requirements for low-carbon development in Sichuan Province.

First, the government should improve the top-level design further and effectively plan for energy conservation and emission reduction. Sichuan Province should dovetail with the national development strategy, listen to different opinions from various parties, formulate a scientific energy conservation and emission reduction plan, and establish an energy planning supervision system to ensure the plan's implementation.

Second, Sichuan Province can fully play a role in the incentives and constraints of finance and taxation. Fiscal subsidies or tax concessions should be given to enterprises that use recycled and alternative resources, meet energy-saving and emission reduction targets, or carry out energy-saving renovations; heavy taxes should be levied on high energy-consuming and high-polluting industries to achieve a reverse reduction in production costs and guide enterprises to follow the path of a low-carbon economy.

Third, the government should deepen the monitoring and accountability mechanism while implementing supervision and inspection of energy-saving and emission-reduction work. Establish a long-term effective monitoring mechanism, investigate and publish the list of illegal units to ensure the effective implementation of emission reduction laws and regulations.

Finally, Sichuan Province should not only establish a platform for sharing information and technology on energy conservation but also rely on existing innovation platforms such as the Chengdu-Chongqing Comprehensive Science Centre and the Western (Chengdu) Science City to speed up vital standard technologies and steadily improve energy processing and energy use efficiency.

## 4.4 Scope for future research

Although this study has conducted an in-depth analysis of the relationship between energy consumption, carbon emissions, and economic growth in Sichuan Province using the Tapio decoupling index and the LMDI decomposition method, there are certain limitations to the research. Firstly, the temporal and geographical scope of the study restricts its universality and currency. Secondly, while the methodologies employed are effective, they have inherent limitations that may not fully reveal the more complex dynamics of economic, social, and technological changes. Therefore, future research needs to expand the temporal and spatial scope, employ more diverse methodologies, and pay closer attention to the role of policy and technological progress to more comprehensively understand and explain the driving factors behind energy consumption and carbon emissions.

## 5. Conclusions

This study delves into the necessity of a low-carbon economy for sustainable social development, with a particular focus on Sichuan Province in China. By utilizing Tapio decoupling indicators and LMDI analysis, the research reveals a relative decoupling relationship between economic growth and energy consumption as well as carbon emissions in Sichuan Province. In Sichuan Province, from 2013 to 2018, there was a decoupling trend observed between energy consumption, carbon emissions, and economic growth, with slower growth in energy use and carbon emissions having a lesser impact on the overall decoupling process. The key decoupling factors influencing the yearly changes in energy consumption in Sichuan Province are population and economic growth, while industrial structure and energy intensity act as crucial limiting factors. The decline in energy intensity negatively affects total energy consumption growth, whereas population and economic growth positively drive stable energy demand growth. Changes in carbon dioxide emissions drivers in Sichuan Province have been analyzed, with economic growth and population abundance playing positive roles, and energy intensity acting as a significant limiting factor. Industrial structure has shown a moderating

effect on carbon emissions growth, particularly after 2011, indicating the impact of modernization and optimization efforts on curbing emissions growth. The paper emphasizes the importance of balancing economic growth with carbon emissions reduction and puts forward targeted policy recommendations to promote coordinated development between the economy and the environment. While the study has made contributions, it also has limitations, such as the need for more in-depth regional analysis and a focus on comprehensive evaluation methods. Future research directions could explore further strategies to achieve sustainable development and address the gaps in understanding the dynamics of decoupling between energy consumption, carbon emissions, and economic growth.

## Supporting information

**S1 Data.**
(XLSX)

## Acknowledgments

The authors would like to express their greatest thanks to the editor and anonymous referees for their helpful comments and suggestions. However, the responsibility for the views expressed as well as any errors or omissions is borne by ours.

## Author Contributions

**Conceptualization:** Genjin Sun.

**Data curation:** Rui Gao, Yanxiu Liu.

**Formal analysis:** Ying Liu.

**Funding acquisition:** Genjin Sun.

**Methodology:** Rui Gao.

**Software:** Rui Gao.

**Supervision:** Ying Liu, Cuilan Li.

**Writing – original draft:** Genjin Sun, Rui Gao, Ying Liu.

**Writing – review & editing:** Genjin Sun, Yanxiu Liu, Cuilan Li.

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
