## [Decision Letter · Decision Letter 0]

29 Jan 2024

PONE-D-23-37224Studying regional low-carbon development: a case study of Sichuan Province in ChinaPLOS ONE

Dear Dr. Sun,

Thank you for submitting your manuscript to PLOS ONE. After careful consideration, we feel that it has merit but does not fully meet PLOS ONE’s publication criteria as it currently stands. Therefore, we invite you to submit a revised version of the manuscript that addresses the points raised during the review process.

We look forward to receiving your revised manuscript.

Kind regards,

Mahdi Moudi, PhD

Academic Editor

PLOS ONE

Journal Requirements:

This work was supported by National Social Science Foundation of China (No. 20XJY005); Research Center of Sichuan Tourism Development, a Key Research Base of Philosophy and Social Sciences in Sichuan Province (No. LY21-03); Li Bing Research Center, which is the Key Research Base of Sichuan Social Science (No. LBYJ2021-006); Research Center of Tuojiang River Basin High-quality Development, a Key Research Base of Social Sciences in Sichuan Province (No. TJGZL2021-14), Research Center of Sichuan County Economic Development, a Key Research Base of Social Sciences in Sichuan Province (No.XY2021002).

This work was supported by National Social Science Foundation of China (No. 20XJY005); Research Center of Sichuan Tourism Development, a Key Research Base of Philosophy and Social Sciences in Sichuan Province (No. LY21-03); Li Bing Research Center, which is the Key Research Base of Sichuan Social Science (No.LBYJ2021-006); Research Center of Tuojiang River Basin High-quality Development, a Key Research Base of Social Sciences in Sichuan Province (No. TJGZL2021-14), Research Center of Sichuan County Economic Development, a Key Research Base of Social Sciences in Sichuan Province (No.XY2021002).The authors would also like to express their greatest thanks to the editor and anonymous referees for their helpful comments and suggestions. However, the responsibility for the views expressed as well as any errors or omissions is borne by ours.

This work was supported by National Social Science Foundation of China (No. 20XJY005); Research Center of Sichuan Tourism Development, a Key Research Base of Philosophy and Social Sciences in Sichuan Province (No. LY21-03); Li Bing Research Center, which is the Key Research Base of Sichuan Social Science (No. LBYJ2021-006); Research Center of Tuojiang River Basin High-quality Development, a Key Research Base of Social Sciences in Sichuan Province (No. TJGZL2021-14), Research Center of Sichuan County Economic Development, a Key Research Base of Social Sciences in Sichuan Province (No.XY2021002).

5. We note that your Data Availability Statement is currently as follows: All relevant data are within the manuscript and its Supporting Information files.

6. Please amend either the abstract on the online submission form (via Edit Submission) or the abstract in the manuscript so that they are identical.

7. We note that Figure 1 in your submission contain [map/satellite] images which may be copyrighted. All PLOS content is published under the Creative Commons Attribution License (CC BY 4.0), which means that the manuscript, images, and Supporting Information files will be freely available online, and any third party is permitted to access, download, copy, distribute, and use these materials in any way, even commercially, with proper attribution. For these reasons, we cannot publish previously copyrighted maps or satellite images created using proprietary data, such as Google software (Google Maps, Street View, and Earth). For more information, see our copyright guidelines: http://journals.plos.org/plosone/s/licenses-and-copyright.

We require you to either present written permission from the copyright holder to publish these figures specifically under the CC BY 4.0 license, or remove the figures from your submission:

Reviewers' comments:

Reviewer's Responses to Questions

**Comments to the Author**

1. Is the manuscript technically sound, and do the data support the conclusions?

Reviewer #1: Yes

Reviewer #2: Yes

Reviewer #3: Yes

2. Has the statistical analysis been performed appropriately and rigorously? 

Reviewer #1: Yes

Reviewer #2: No

Reviewer #3: No

3. Have the authors made all data underlying the findings in their manuscript fully available?

Reviewer #1: Yes

Reviewer #2: Yes

Reviewer #3: No

4. Is the manuscript presented in an intelligible fashion and written in standard English?

Reviewer #1: Yes

Reviewer #2: No

Reviewer #3: Yes

5. Review Comments to the Author

Reviewer #1: I have reviewed the manuscript entitled “ Studying regional low-carbon development: a case study of Sichuan Province in China”. The manuscript investigates low-carbon development in Sichuan, China, analyzing the relationship between energy consumption, carbon emissions, and economic growth. Using the Tapio decoupling indicator and LMDI, it reveals a relative decoupling in the region and identifies population and economic growth as key drivers, with energy intensity and industrial structure acting as constraints.

The manuscript's topic is interesting, well-written, and provides a valuable reference for the field. However, before publication, it needs some modification. Therefore, I recommend revisions, with below specific comments:

State of art shall be mentioned more clearly

Practical utilization should be discussed further at the end of discussion section

Limitation of the research shall be acknowledged

Update introduction get insight with the available resources and perspective of urban climates for example: https://doi.org/10.1016/j.scs.2023.104459 ; Urban Climate 51, 101637 ;

Also Consider incorporating insights from sources https://doi.org/10.17660/ActaHortic.2023.1374.14 and https://doi.org/10.17660/ActaHortic.2023.1374.11 ; Land 12 (8), 1623 ; to acknowledge the various benefits of green spaces relevant to this study.

Improve the figures quality e.g., figure 5,7,…

Number of references is low;

Re-organize the manuscript. I suggest 1. Introduction 2 methods 3 results 4 discussion (include practical utilization and limitations) 5 conclusion

Reviewer #2: The manuscript attempts to examine the relationship between energy consumption, carbon emissions and economic growth using a variety of statistical methods. The paper is well structured and methodologically sound. However, it lacks innovation in some aspects. Therefore, it is necessary for the manuscript to be improved before publication, especially in the introduction, discussion and practical application sections of the paper. I suggest a major revision.

Specific comments:

(1) Abstract section: Lines 16 to 19. it is difficult to understand the status in different periods via the expression. Please do not express such results in subordinate clauses

(2) In the introduction section, the authors describe a lot of research background, especially the presentation of some data. However, I would have preferred to see the significance of the study, such as the authors' statement that it is the first study and fill a research gap.

(3) Line 88. what does OECD mean?

(4) All the tables and figures, please add the units if it is needed.

(5) Discussion section, I would like to see the theoretical implications of the paper. For instance, comparisons and enhancements with related studies.

(6) This paper has yet to be polished.

(7) Line 272. It is strange that Fig.2 does not show any GDP information. I find that in Figure 3, but the most key point is, why the GDP curve has a great decrease in 2020? it is so strange. please also check the curve of cabon emission.

Reviewer #3: The manuscript needs major revision due to its uncategorized and insufficient Introduction and Method section. Introduction needs more clear notice regarding the topic. Also, Method could be presented property.

6. PLOS authors have the option to publish the peer review history of their article (what does this mean?). If published, this will include your full peer review and any attached files.

Reviewer #1: **Yes: **Majid Amanibeni

Reviewer #2: No

Reviewer #3: No

---

## [Author Response · Author response to Decision Letter 0]

6 Mar 2024

Response to Reviewers’ Comments

We appreciate very much for your decision and constructive comments on our manuscript. These comments are very helpful for us to revise this paper. We have carefully considered the suggestions of the Reviewers, and tried our best to carry out a comprehensive revision to improve the manuscript. The revision notes are addressed point by point below.

Response to Reviewer 1 Comments

Point 1: State of art shall be mentioned more clearly

Response 1: Thank you for your careful review and valuable suggestion. We have revised the manuscript to provide a clearer and more comprehensive overview of the state of the art in the relevant field. Specifically, we have summarized the current mainstream viewpoints and research directions in the field, while also highlighting the existing shortcomings and unresolved issues. Additionally, we have explicitly stated the specific research questions and objectives that our study addresses, aiming to provide readers with a clearer understanding of the background and positioning of our research. The above-mentioned modifications are available in Lines 75-142 in Section 1.

Point 2: Practical utilization should be discussed further at the end of the discussion section

Response 2: Thank you very much for your suggestion. We have adjusted the section “5.2 Policy Implications” to “4.3 Policy Implications”, and added a dedicated paragraph that emphasizes the potential application prospects and the inherent value of our research outcomes (Lines 427-458). This includes discussions on how to address the unresolved issues that future research should focus on. Furthermore, we have highlighted how our research can contribute to the advancement of the field.

Point 3: Limitation of the research shall be acknowledged

Response 3: Thank you very much for your suggestion. In response to this comment, we have included the section “4.4 Scope for Future Research”, available in Lines 459-468 in the Manuscript. In the section, we explicitly acknowledged the limitations and shortcomings of our research. This includes discussing the limitations related to sample sizes and the potential shortcomings of the research methods employed, the simplifications in research assumptions, and the scope limitations. By addressing these limitations, we aim to provide a balanced and transparent presentation of our study.

Point 4: Update introduction get insight with the available resources and perspective of urban climates for example: https://doi.org/10.1016/j.scs.2023.104459 ; Urban Climate 51, 101637 ;Also Consider incorporating insights from sources https://doi.org/10.17660/ActaHortic.2023.1374.14 and https://doi.org/10.17660/ActaHortic.2023.1374.11 ; Land 12 (8), 1623 ;to acknowledge the various benefits of green spaces relevant to this study.

Response 4: Thank you for the valuable references. We have updated and enriched the introduction section by incorporating insights from the provided references, available in Lines 35, 37 in the Manuscript. Specifically, we have integrated perspectives from the referenced papers to enhance the discussion of the current research progress and viewpoints in the field of urban climates. Additionally, we have incorporated the various benefits of green spaces relevant to our study, such as temperature reduction and pollution absorption, based on the insights from the provided sources. We have also updated the reference section to include these additional sources, thereby enriching the background of our research.

Point 5: Improve the figures quality e.g., figure 5,7,…

Response 5: Thank you very much for your suggestion, we have made the following changes to address the issue of chart quality, firstly, to reduce the problem of image clarity degradation during file transfer, we have updated all charts to the Scalable Vector Graphics format. Secondly, to improve the aesthetics and clarity of the charts, we have adjusted the colour scheme of the charts.

Point 6: Number of references is low.

Response 6: Thank you for your suggestions. To make the paper more in-depth and informative, we have added other 12 citations. In total, there are 51 citations throughout the paper. By adding these citations, we have compared the content of this paper with previous research, thus increasing the scientific validity and credibility of our study.

Point 7: Re-organize the manuscript. I suggest 1. Introduction 2 methods 3 results 4 discussion (include practical utilization and limitations) 5 conclusion

Response 7: We are grateful for the suggestion. To make the paper more rigorous, it has now been structured as follows: Section 1 Introduction, Section 2 Methods, Section 3 Results, Section 4 Discussions, and Section 5 Conclusions.

Response to Reviewer 2 Comments

Point 1: Abstract section: Lines 16 to 19. it is difficult to understand the status in different periods via the expression. Please do not express such results in subordinate clauses.

Response 1: We appreciate the feedback regarding the clarity of expression in the abstract section. In response to this, we have revised the mentioned lines to improve the comprehensibility of the status in different periods. Specifically, we have extracted the content from the subordinate clause and presented it as standalone sentences. By directly stating the research status in different periods and avoiding complex grammatical structures, we aim to enhance the clarity and ease of understanding for readers.

Point 2: In the introduction section, the authors describe a lot of research background, especially the presentation of some data. However, I would have preferred to see the significance of the study, such as the authors' statement that it is the first study and fill a research gap.

Response 2: Thank you for your careful review and valuable suggestion. We have restructured and enriched the introduction section to address the significance of the study as suggested. Specifically, we have included a concise description of the necessary research background knowledge while emphasizing the originality and innovation of our study . Additionally, we have explicitly stated the significance and contributions of our research, including how it fills existing research gaps. We have clarified the research objectives and tasks, and we have utilized data to support our points effectively. Furthermore, we have highlighted the potential impact of our research on both academic and practical domains.The above-mentioned modifications are available in Lines 114-142 in Section 1.

Point 3: Line 88. what does OECD mean?

Response 3: Thank you for your careful review. OECD is the abbreviation of the Organization for Economic Co-operation and Development. While keeping the language simple and fluent, we have made the necessary changes to the abbreviation with its full name. In addition, we checked other abbreviations in the paper to ensure that they were accompanied by a clear explanation the first time they appeared.

Point 4: All the tables and figures, please add the units if it is needed.

Response 4: We think this is an excellent suggestion. We checked the images and tables in the paper and labelled the data that required the use of units with the appropriate units as a way of presenting our findings more accurately.

Point 5: Discussion section, I would like to see the theoretical implications of the paper. For instance, comparisons and enhancements with related studies.

Response 5: Thank you for your suggestions. In order to illustrate the practical and theoretical significance of this paper, we have added the related contents in the Discussion section (Lines 431-436). Specifically, in terms of theoretical significance, this study enriches the research content of regional economics by illustrating the intricate and dynamic relationship between economic growth, energy consumption and carbon emissions through a detailed case study of Sichuan Province. In addition, this study contributes to a deeper understanding of the mechanisms behind the optimisation of energy efficiency and industrial structure, and provides valuable insights for the formulation of sustainable development policies.

Point 6: This paper has yet to be polished.

Response 6: Thank you for your suggestions. After fully considering the valuable comments of the reviewers, we have revised the paper carefully, by checking the grammar, modifying appropriate words, optimizing sentence structures and so on. 

Point 7: Line 272. It is strange that Fig.2 does not show any GDP information. I find that in Figure 3, but the most key point is, why the GDP curve has a great decrease in 2020? it is so strange. please also check the curve of carbon emission.

Response 7: Thank you for your careful review. We have checked the content of Figures 2 and 3 and found that the statement at Line 272 was a chart labelling error and we have corrected it accordingly (Line 307, the new Fig. 2 in Lines 72-74). Regarding your suggestion to check the GDP data, we are very concerned about it. After checking, we found that it was a data editing error in the process of making the charts, and we have corrected the GDP data in Fig. 2. In order to avoid the a error, we have rechecked and corrected all the data in the full text.

Response to Reviewer 3 Comments

Point 1: The manuscript needs major revision due to its uncategorized and insufficient Introduction and Method section. Introduction needs more clear notice regarding the topic.

Response 1: Thank you for your review and constructive comments on our manuscript. We appreciate the suggestions you made to strengthen key sections. In response to point 1 about the need for a clearer introduction to the topic, we have thoroughly revised the introduction section. We clearly stated the research objectives and questions. We also provided a detailed review of relevant theories and research and clearly articulated the innovations of the study.

We also revised the methodology section. To make the use of methods clearer and more explicit, we added empirical thinking (Lines 149-155). This section describes which elements we primarily use TAPIO and LMDI to demonstrate. In addition, we have adjusted the language of the methodology section so that our use of the methodology can be expressed more clearly.

---

## [Decision Letter · Decision Letter 1]

1 Apr 2024

PONE-D-23-37224R1Studying regional low-carbon development: a case study of Sichuan Province in ChinaPLOS ONE

Dear Dr. Sun,

Thank you for submitting your manuscript to PLOS ONE. After careful consideration, we feel that it has merit but does not fully meet PLOS ONE’s publication criteria as it currently stands. Therefore, we invite you to submit a revised version of the manuscript that addresses the points raised during the review process.

We look forward to receiving your revised manuscript.

Kind regards,

Mahdi Moudi, PhD

Academic Editor

PLOS ONE

Journal Requirements:

Reviewers' comments:

Reviewer's Responses to Questions

**Comments to the Author**

1. If the authors have adequately addressed your comments raised in a previous round of review and you feel that this manuscript is now acceptable for publication, you may indicate that here to bypass the “Comments to the Author” section, enter your conflict of interest statement in the “Confidential to Editor” section, and submit your "Accept" recommendation.

Reviewer #1: All comments have been addressed

2. Is the manuscript technically sound, and do the data support the conclusions?

Reviewer #1: Yes

3. Has the statistical analysis been performed appropriately and rigorously? 

Reviewer #1: Yes

4. Have the authors made all data underlying the findings in their manuscript fully available?

Reviewer #1: No

5. Is the manuscript presented in an intelligible fashion and written in standard English?

Reviewer #1: Yes

6. Review Comments to the Author

Reviewer #1: The manuscript has been improved after the first round of revision. However, there is still room for further improvement:

- The conclusion should be shorter, preferably one paragraph, concise, and depict the broader message of the study.

7. PLOS authors have the option to publish the peer review history of their article (what does this mean?). If published, this will include your full peer review and any attached files.

Reviewer #1: No

---

## [Author Response · Author response to Decision Letter 1]

8 Apr 2024

We would like to thank you for your professional review work, constructive comments, and valuable suggestions on our manuscript. Your time and efforts are greatly appreciated. We have strived to improve the manuscript accordingly as listed in details below.

Point 1: The conclusion should be shorter, preferably one paragraph, concise, and depict the broader message of the study.

Response 1: Thanks to your careful review and valuable suggestions, we have streamlined the conclusion section of the article. Mainly, we have consolidated the expression of 469-495 lines in the previous draft into a one-paragraph expression of 469-489 lines, and the word count has been streamlined from 362 words to the current 283 words. Considering the content of the conclusion section needs to depict the broader message. We conclude with a brief summary of the paper’s findings, research contributions, research limitations, and suggestions for future research directions. Due to the large amount of information that needs to be presented, it has been streamlined as much as possible, leaving the content that is important and needs to be explained in the conclusion.

In addition, we have rechecked the manuscript again and improved some details, available in Revised the Manuscript with Tracked Changes-PONE-D-23-37224.

---

## [Decision Letter · Decision Letter 2]

11 Apr 2024

Studying regional low-carbon development: a case study of Sichuan Province in China

PONE-D-23-37224R2

Dear Dr. Sun,

We’re pleased to inform you that your manuscript has been judged scientifically suitable for publication and will be formally accepted for publication once it meets all outstanding technical requirements.

Kind regards,

Mahdi Moudi, PhD

Academic Editor

PLOS ONE

Additional Editor Comments (optional):

Reviewers' comments:

Reviewer's Responses to Questions

**Comments to the Author**

1. If the authors have adequately addressed your comments raised in a previous round of review and you feel that this manuscript is now acceptable for publication, you may indicate that here to bypass the “Comments to the Author” section, enter your conflict of interest statement in the “Confidential to Editor” section, and submit your "Accept" recommendation.

Reviewer #1: All comments have been addressed

2. Is the manuscript technically sound, and do the data support the conclusions?

Reviewer #1: Yes

3. Has the statistical analysis been performed appropriately and rigorously? 

Reviewer #1: Yes

4. Have the authors made all data underlying the findings in their manuscript fully available?

Reviewer #1: Yes

5. Is the manuscript presented in an intelligible fashion and written in standard English?

Reviewer #1: Yes

6. Review Comments to the Author

Reviewer #1: the manuscript is acceptable as the authors have incorporated with the comments and the manuscript has been improved significantly

7. PLOS authors have the option to publish the peer review history of their article (what does this mean?). If published, this will include your full peer review and any attached files.

Reviewer #1: No

---

## [Editor Report · Acceptance letter]

29 Apr 2024

PONE-D-23-37224R2 

PLOS ONE

Dear Dr. Sun, 

I'm pleased to inform you that your manuscript has been deemed suitable for publication in PLOS ONE. Congratulations! Your manuscript is now being handed over to our production team.

Kind regards, 

on behalf of

Dr. Mahdi Moudi 

Academic Editor

PLOS ONE